 SHORT REPORT

# Conscious processing of global and local auditory irregularities causes differentiated heartbeat-evoked responses

Diego Candia-Rivera[1,2]*, Federico Raimondo[2,3,4], Pauline Pérez[2,5], Lionel Naccache[2,6,7,8,9], Catherine Tallon-Baudry[1], Jacobo D Sitt[2,7]*

[1]Laboratoire de Neurosciences Cognitives et Computationnelles, Département d'Etudes Cognitives, École Normale Supérieure, INSERM, Université PSL, Paris, France; [2]Sorbonne Université, Paris Brain Institute (ICM), INRIA, CNRS, INSERM, AP-HP, Hôpital Pitié-Salpêtrière, Paris, France; [3]Institute of Neuroscience and Medicine (INM-7: Brain and Behaviour), Forschungszentrum Jülich, Jülich, Germany; [4]Institute of Systems Neuroscience, Heinrich Heine University Düsseldorf, Düsseldorf, Germany; [5]AP-HP, Hôpital de la Pitié Salpêtrière, Neuro ICU, DMU Neurosciences, Paris, France; [6]Pitié-Salpêtrière Faculty of Medicine, Pierre and Marie Curie University, Sorbonne Universities, Paris, France; [7]INSERM, National Institute of Health and Medical Research, Paris, France; [8]Department of Neurology, Pitié-Salpêtrière Hospital Group, Public Hospital Network of Paris, Paris, France; [9]Department of Neurophysiology, Pitié-Salpêtrière Hospital Group, Public Hospital Network of Paris, Paris, France

*For correspondence:
diego.candia.r@ug.uchile.cl
(DC-R);
jacobo.sitt@icm-institute.org
(JDS)

**Competing interest:** The authors declare that no competing interests exist.

**Abstract** Recent research suggests that brain-heart interactions are associated with perceptual and self-consciousness. In this line, the neural responses to visceral inputs have been hypothesized to play a leading role in shaping our subjective experience. This study aims to investigate whether the contextual processing of auditory irregularities modulates both direct neuronal responses to the auditory stimuli (ERPs) and the neural responses to heartbeats, as measured with heartbeat-evoked responses (HERs). HERs were computed in patients with disorders of consciousness, diagnosed with a minimally conscious state or unresponsive wakefulness syndrome. We tested whether HERs reflect conscious auditory perception, which can potentially provide additional information for the consciousness diagnosis. EEG recordings were taken during the local-global paradigm, which evaluates the capacity of a patient to detect the appearance of auditory irregularities at local (short-term) and global (long-term) levels. The results show that local and global effects produce distinct ERPs and HERs, which can help distinguish between the minimally conscious state and unresponsive wakefulness syndrome patients. Furthermore, we found that ERP and HER responses were not correlated suggesting that independent neuronal mechanisms are behind them. These findings suggest that HER modulations in response to auditory irregularities, especially local irregularities, may be used as a novel neural marker of consciousness and may aid in the bedside diagnosis of disorders of consciousness with a more cost-effective option than neuroimaging methods.

## Editor's evaluation

This study shows that neural responses to sounds and to heartbeats are affected in different ways by short-term and long-term auditory irregularities in patients diagnosed with a minimally conscious

state or unresponsive wakefulness syndrome. While the findings would have been more robust had the authors collected data in the same way from a larger group of control subjects, they highlight the potential value of using heartbeat-evoked responses to inform the bedside diagnosis of disorders of consciousness. More generally, they will of interest to researchers studying brain-body interactions and their relationship to perceptual awareness.

## Introduction

Theoretical developments in consciousness and experimental research have rooted the basis of consciousness in how the brain responds to visceral inputs (*Azzalini et al., 2019*; *Candia-Rivera, 2022a*; *Park and Tallon-Baudry, 2014*). In post-comatose patients, the consciousness diagnosis is primarily based on behavioral signs of consciousness (*Bayne et al., 2017*), which aims at distinguishing between patients showing only reflex-like responses to the environment, diagnosed as Vegetative State or Unresponsive Wakefulness Syndrome (VS/UWS; *Laureys et al., 2010*), and patients with fluctuating but reproducible signs of non-reflex behavior, diagnosed as a Minimally Conscious State (MCS), (*Giacino et al., 2002*), but see also *Naccache, 2018*. However, recent results demonstrate that behavioral assessment is not sufficient and neuroimaging techniques are used to detect covert states of consciousness (*Kondziella et al., 2020*).

The classification of MCS and UWS patients using EEG and cardiac features while undergoing processing of auditory regularities has shown an advantage over EEG features alone (*Raimondo et al., 2017*), implying that brain-heart interactions may be involved in the conscious processing of auditory inputs. Recent evidence on automatic classifications of HERs in the resting-state showed that these markers may capture residual signs of consciousness (*Candia-Rivera et al., 2021a*; *Candia-Rivera and Machado, 2023a*) suggesting that HERs might convey state-of-consciousness relevant information about how the brain responds to bodily-related stimuli. Further evidence exists in healthy participants, in which the processing of auditory stimuli may cause cognitive modulations to the cardiac cycle (*Tanaka et al., 2023*; *Marshall et al., 2022*; *Banellis and Cruse, 2020*; *Perez et al., 2020*; *Pfeiffer and De Lucia, 2017*), and HERs correlate with perceptual awareness (*Al et al., 2020*; *Banellis and Cruse, 2020*; *Park et al., 2014*).

We hypothesized that HERs can be modulated by contextual processing of different levels of auditory regularities, as presented in the local-global paradigm (*Bekinschtein et al., 2009*). In this study, we analyze HERs following the presentation of auditory irregularities, with special regard for distinguishing UWS (n=40) and MCS (n=46) patients. Note that the automated classification of this cohort was previously performed in another study (*Raimondo et al., 2017*). Therefore, our aim is to characterize the group-wise differences between UWS and MCS patients that may allow a multi-dimensional cognitive evaluation to infer the presence of consciousness (*Sergent et al., 2017*), but

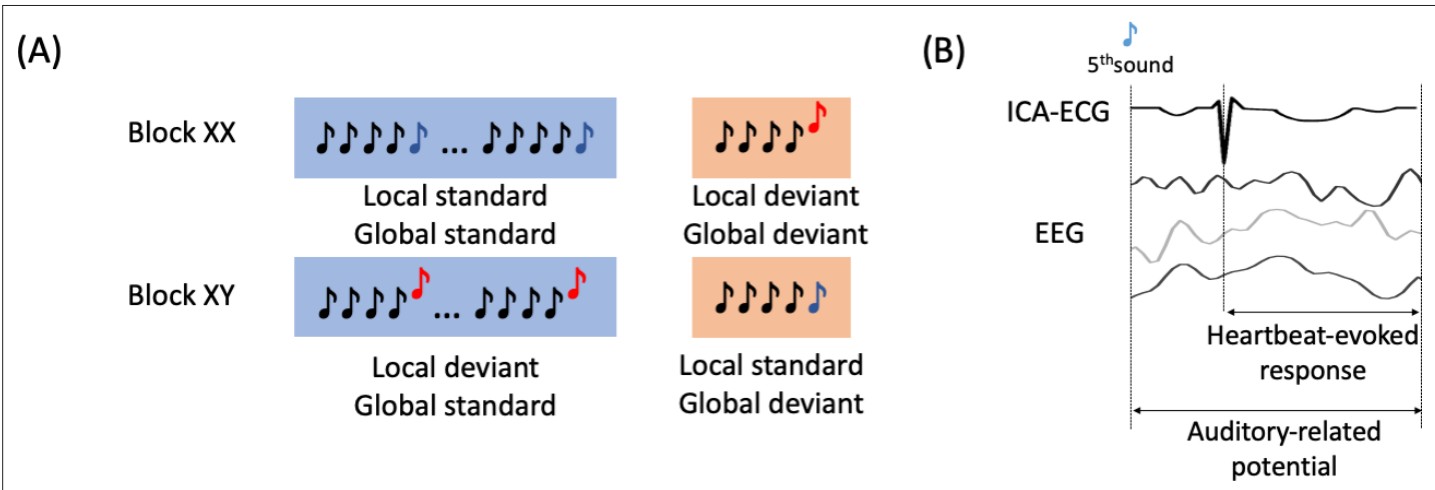

**Figure 1.** Experimental description and EEG analysis. (**A**) Local-global paradigm. (**B**) Heartbeat-evoked response defined by the R-peaks that follow the 5th sound from all the trials, and the Auditory-related potential defined by the EEG activity locked to the stimuli.

also complement the bedside diagnosis performed with neuroimaging methods that capture neural correlates of covert consciousness (*Sanz et al., 2021*).

## Results

This study employed high-density EEG recordings to assess the cognitive processing of auditory irregularities in patients with disorders of consciousness using the local-global paradigm (*Bekinschtein et al., 2009*). This paradigm evaluates auditory regularities at both short-term (local) and long-term (global) levels within trials of five consecutive sounds. The 5th sound distinguishes standard from deviant trials at both local and global levels. As depicted in *Figure 1A*, XX, and XY types of blocks were presented. In XX blocks, frequent stimuli consisted of five equal sounds (local and global standard), whereas infrequent stimuli had four equal sounds followed by a different 5th sound (local and global deviant). In XY blocks, frequent stimuli involved four equal sounds followed by a different 5th sound (local deviant and global standard), while infrequent stimuli featured five equal sounds (local standard and global deviant). We examined the cognitive processing of auditory irregularities with the objective to identify the physiological responses that could differentiate between patients in MCS and those in UWS. We hypothesized that assessing auditory irregularities at both local and global levels could offer valuable insights into the distinction of MCS and UWS patients. Furthermore, that distinction may be further improved by analyzing the physiological modulation of auditory processing in relation to measures of brain-heart interactions. To achieve this, we conducted tests to investigate the local and global effects of ERPs, which involved analyzing the standard average of EEG epochs aligned with the occurrence of auditory deviants (*Figure 1B*). Additionally, we explored the HERs, which involved analyzing the average of EEG epochs aligned with the occurrence of heartbeats following the auditory deviants (*Figure 1B*). We aimed to assess whether the neural responses to heartbeats, within the context of auditory irregularity processing, could serve as novel differentiating factors between MCS and UWS patients.

First, unpaired non-parametric cluster analysis was performed between MCS and UWS patients for ERPs, global and local effects. The local effect involved calculating the average of EEG epochs associated with local deviants (comprising local deviant/global standard epochs and local deviant/global deviant epochs) and subtracting the average of the EEG epochs associated with local standards (comprising local standard/global standard epochs and local standard/global deviant epochs). The global effect involved calculating the average of EEG epochs linked to global deviants (comprising local standard/global deviant epochs and local deviant/global deviant epochs) and subtracting the average of EEG epochs associated with global standards (comprising local standard/global standard epochs and local deviant/global standard epochs). *Figure 2A* shows the clustered effects found with respect to the 5th sound, in the ERP global effect (main positive cluster: p=0.0001, Z=3.684, latency = 800–850 ms; main negative cluster: p=0.0013, Z=–3.1905, latency = 280–336 ms) and ERP local effect (main positive cluster: p=0.0011, Z=3.4416, latency = 236–328 ms). The clustered effects were combined to obtain a single value for each patient, corresponding to ERP global and local effects. To combine the clustered effects, we computed the average of all points (channel × time) identified in the cluster permutation analysis, which effectively distinguished between patients diagnosed with MCS and UWS. The distribution of the combined clustered effects are depicted in *Figure 2B* and the time course of one of the channels of the cluster in *Figure 2C*.

Consecutively, cluster permutation analysis was performed between MCS and UWS patients for HERs, global and local effects. In *Figure 3A* are shown the clustered effects found with respect to the R-peak following the 5th sound, in the HER global effect (main positive cluster: p=0.0037, Z=3.0173, latency = 112–130 ms; main negative cluster: p0.0058, Z=–3.0173, latency = 340–360 ms) and HER local effect (main positive cluster: p=0.0029, Z=3.0606, latency = 400–412 ms; main negative cluster: p=0.0014, Z=–3.3983, latency = 0–40 ms). The clustered effects were combined to obtain a single value for each patient, corresponding to HER global and local effects. The distribution of the combined clustered effects are depicted in *Figure 3B* and the time course of one of the channels of the cluster in *Figure 3C*. The combined clustered effects were compared to 100 randomly distributed heartbeats to compute the surrogate p-value. The HER local effect was larger than what would be expected by chance as estimated from surrogate heartbeats (HER local effect, Monte Carlo p=0.03; HER global effect, Monte Carlo p=0.54).

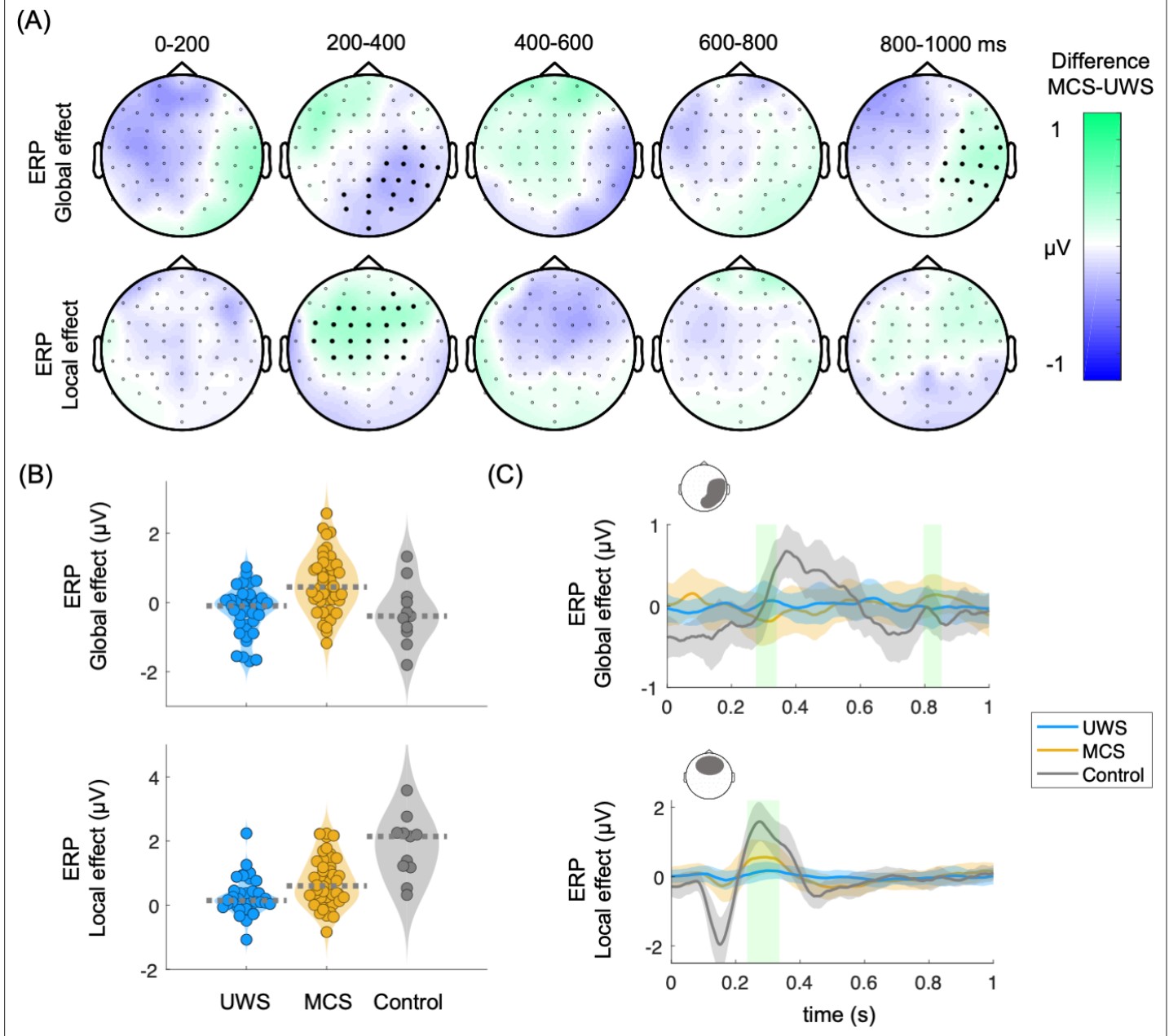

**Figure 2.** Auditory event-related potentials (ERPs) in the global and local effects. (**A**) Scalp topographies indicate the average group differences between MCS and UWS patients. Thick electrodes indicate a clustered effect (Monte Carlo p<0.05). (**B**) Average of the clustered effects per patient, in the ERP global effect (main positive cluster: p=0.0001, Z=3.684, latency = 800–850 ms; main negative cluster: p=0.0013, Z=–3.1905, latency = 280–336 ms), and ERP local effect (main positive cluster: p=0.0011, Z=3.4416, latency = 236–328 ms). Healthy controls are displayed as a reference. Dashed lines indicate the group median (**C**) Time course of the group median among UWS, MCS, and control groups. The displayed time course corresponds to the scalp area marked above the corresponding plot. Shaded green areas indicate the segments in which a clustered effect was found when comparing MCS and UWS groups. ERPs: auditory event-related potentials, MCS: minimally conscious state, UWS: unresponsive wakefulness syndrome.

We then tested whether the clusters found using cluster permutations at global and local effects, as measured from HERs and ERPs, come from a distribution with a median different from zero, i.e., whether the deviants differ from the standard 5th sounds within patients' groups (*Table 1*). We found a significant ERP and HER local effect in both MCS and UWS patients. On the other hand, the global effect was significant only for MCS patients in both ERP and HER analysis. This result extends previous reports highlighting the predictive power for the conscious state of the global effect (*Pérez et al., 2021*).

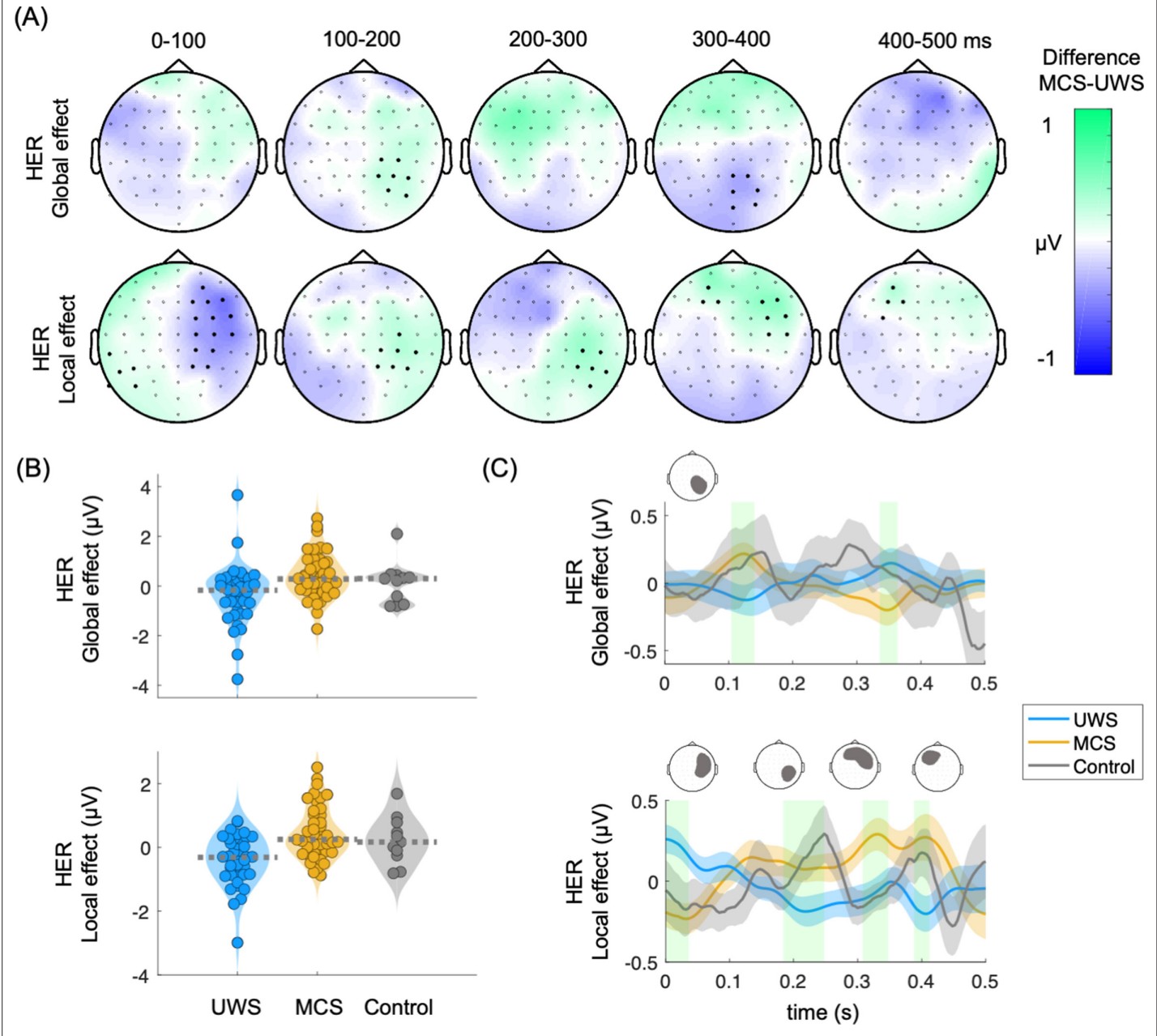

**Figure 3.** Heartbeat-evoked responses (HERs) in the global and local effects. (**A**) Scalp topographies indicate the average group differences between MCS and UWS patients. Thick electrodes indicate a clustered effect (Monte Carlo p<0.05). (**B**) Average of the clustered effects per patient, in the HER global effect (main positive cluster: p=0.0037, Z=3.0173, latency = 112–130 ms; main negative cluster: p=0.0058, Z=−3.0173, latency = 340–360 ms) and HER local effect (main positive cluster: p=0.0029, Z=3.0606, latency = 400–412 ms; main negative cluster: p=0.0014, Z=−3.3983, latency = 0–40 ms). Healthy controls are displayed as a reference. Dashed lines indicate the group median (**C**) Time course of the group median among UWS, MCS, and control groups. The displayed time course corresponds to the scalp area marked above the corresponding plot. Shaded green areas indicate the segments in which a clustered effect was found when comparing MCS and UWS groups. HERs: heartbeat-evoked responses, MCS: minimally conscious state, UWS: unresponsive wakefulness syndrome.

In *Figure 4A* are presented all pair comparisons between ERPs and HERs. for local and global effects. The figure depicts that the measured effects do not show apparent correlations (details on Spearman correlation tests in *Table 2*). *Figure 4B* shows that the four markers: ERP global, ERP local, HER global, and HER local present complementary information for the separation of the diagnostic groups.

**Table 1.** Wilcoxon sign test performed separately for MCS and UWS patients, to test whether the global and local effects as measured from HERs and ERPs come from a distribution with median different to zero.

Bold indicates significance reached at $\alpha$=0.05/8=0.0063, according to Bonferroni correction for multiple comparisons.

| Patients | HERs | | ERPs | |
|---|---|---|---|---|
| | Global effect | Local effect | Global effect | Local effect |
| MCS | Z=2.7805 p=0.0054 | Z=3.2175 p=0.0013 | Z=3.7529 p=0.0002 | Z=5.0311 p<0.0001 |
| UWS | Z=−1.9759 p=0.0482 | Z=−2.9840 p=0.0028 | Z=−1.9624 p=0.0497 | Z=2.9033 p=0.0037 |

HERs: heartbeat-evoked responses, ERPs: auditory event-related potentials, MCS: minimally conscious state, UWS: unresponsive wakefulness syndrome.

To further demonstrate the discrimination power of MCS and UWS patients using HERs and ERPs, we employed a linear discriminant classifier in a fivefold cross-validation. *Figure 4C* illustrates that combining HER local, ERP global, and ERP local offered the most complementary information out of all possible triads, achieving a cross-validation accuracy of 79.1%. The accuracy further improved to 80.2% when incorporating the four features. These findings highlight the additional insights provided by HERs in conjunction with the standard ERP analysis.

HER average during the whole protocol presents a small, clustered effect when comparing MCS and UWS patients (*Figure 5A*, left). In *Figure 5A*, the right panel is shown that a higher HER variance is observed in MCS compared to UWS during the whole protocol. A wide scalp coverage presents higher HER variance in MCS, as compared to UWS (cluster permutation test, p<0.0001, Z=4.0772, latency = 20–500 ms). The time courses of the clustered effects in HER average and variance are shown in *Figure 5B*.

## Discussion

Considering that brain-heart interactions have demonstrated to be involved in consciousness and relevant for the clinical assessment of brain-injured patients (*Candia-Rivera et al., 2021a*; *Candia-Rivera and Machado, 2023b*; *Perez et al., 2020*; *Raimondo et al., 2017*; *Riganello et al., 2019*), we analyzed neural responses to heartbeats during the processing of auditory irregularities to characterize MCS and UWS patients. The processing of short- and long-term auditory irregularities, i.e., the local and global effects, shows distinctive responses between MCS and UWS patients in their HERs.

The correlation analyses revealed that the EEG signals synchronized to heartbeats (HERs) provided complementary information to the ERPs synchronized to auditory irregularities. Examining the local effects using HERs and ERPs yielded better differentiability between MCS and UWS patients (see *Figure 5*).

It is worth noting that the HER local effect demonstrated higher specificity, as compared to the HER global effect during the permutation test. Only 3% of randomly timed surrogate heartbeats exhibited separability that surpassed what was observed with the original heartbeats. These results align with previous findings that suggest the existence of a short-term auditory-cardiac synchrony (*Banellis and Cruse, 2020*; *Pérez et al., 2021*; *Pfeiffer and De Lucia, 2017*). Moreover, our findings indicate that brain-heart dynamics may serve as markers of the conscious processing of auditory information, particularly in distinguishing short-term changes.

Our results go in the same direction as previous evidence, in which automatic classifications of these patients showed a higher accuracy when locking EEG to heartbeats, with respect to the classification of EEG segments unrelated to the cardiac cycle (*Candia-Rivera et al., 2021a*). Nevertheless, the measured responses in ERPs and HERs do not separate MCS and UWS patients' groups completely (see *Figure 4C*), suggesting that some patients do not react or only react to some trials that were attenuated when averaging all trials in the time-locked analysis. Furthermore, it is worth noting that the ERP global effect observed in healthy controls did not follow the same trend as MCS patients. These findings may require of further future explorations to determine if the observed effect

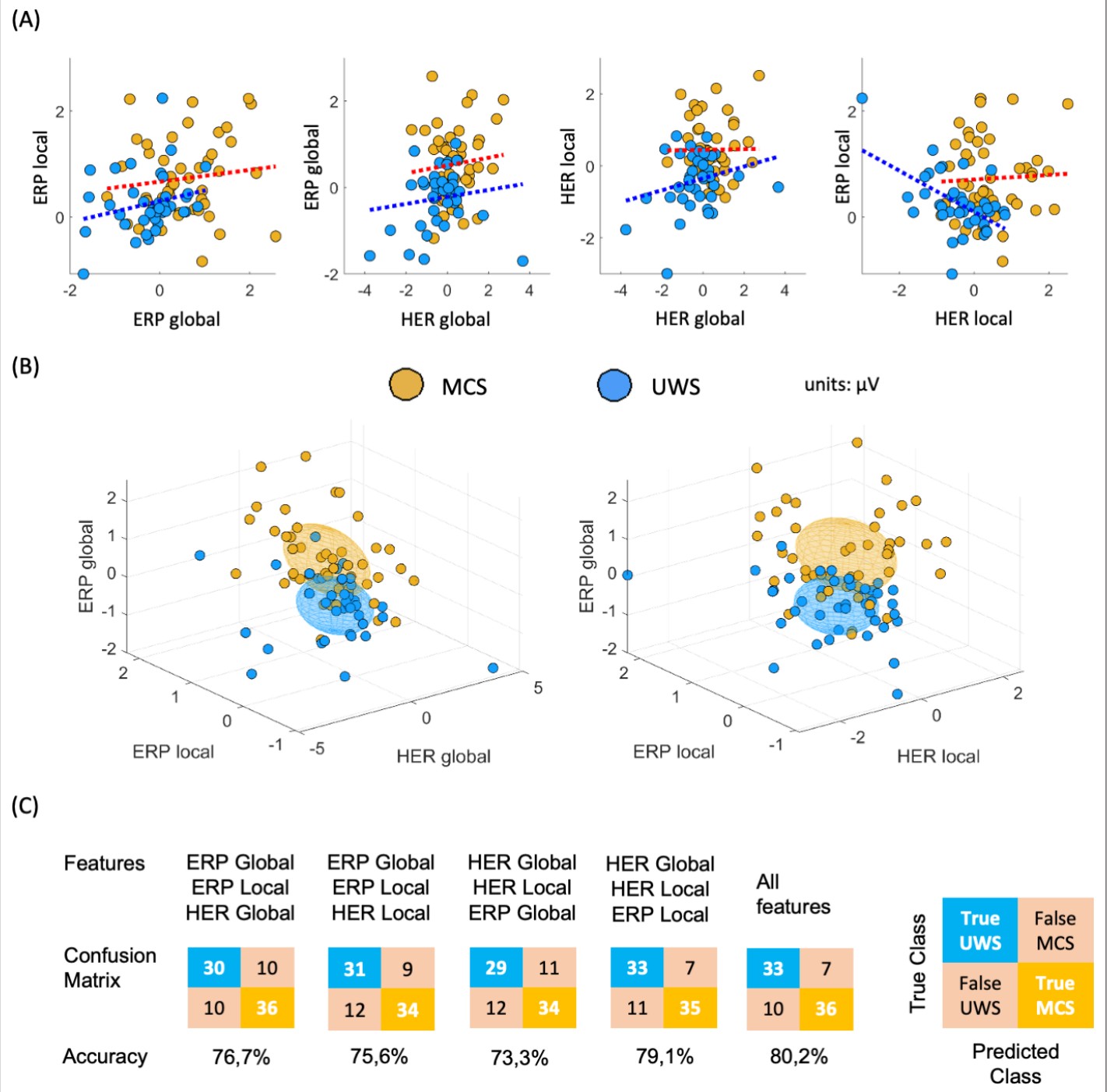

**Figure 4.** Multi-dimensional analysis of the clustered effects found when comparing MCS and UWS patients. (**A**) Pairwise comparison between all possible combinations for ERPs and HERs, for local and global effects. Individual points corresponding to a single patient, and dotted line indicates the trend, separately per diagnosis. (**B**) Three-dimensional representation of the clustered effects: left panel for ERP global, ERP local, and HER global; and right panel for ERP global, ERP local, and HER local. *E*ach ellipsoid was constructed per diagnostic group, centered in the group means with a ratio defined by the standard deviations, for the respective dimensions. (**C**) Confusion matrices depicting the classification results of MCS and UWS patients using a linear discriminant classifier in a fivefold cross-validation. The classifiers were trained using all possible combinations of feature triads, as well as all four features. HERs: heartbeat-evoked responses, ERPs: auditory event-related potentials, MCS: minimally conscious state, UWS: unresponsive wakefulness syndrome.

**Table 2.** Group-wise Spearman correlation analysis performed separately for MCS and UWS patients, between the combined clustered effects found when comparing MCS vs UWS in the ERP global effect, ERP local effect, HER global effect, and HER local effect.

Significance was set at $\alpha$=0.05/8=0.0063, according to Bonferroni correction for multiple comparisons.

|  | MCS | UWS |
|---|---|---|
| ERP global vs ERP local | R=0.1077<br>p=0.4748 | R=0.3099<br>p=0.0591 |
| HER global vs ERP global | R=0.0575<br>p=0.7033 | R=0.1580<br>p=0.3290 |
| HER global vs HER local | R=−0.1193<br>p=0.4283 | R=0.1480<br>p=0.3607 |
| HER local vs ERP local | R=−0.0436<br>p=0.7730 | R=−0.4114<br>p=0.0088 |

HER: heartbeat-evoked response, ERP: auditory event-related potential, MCS: minimally conscious state, UWS: unresponsive wakefulness syndrome.

is exclusive to MCS patients or if healthy controls do not exhibit the same effect due to their lower number of trials performed in the local-global paradigm during this study. However, the response observed in healthy controls does resemble a standard P300 response. These findings align with previous reports indicating that the ERPs during local deviants exhibit superior discriminatory ability between MCS and UWS patients (*Faugeras et al., 2012*). Furthermore, these results suggest that both ERP and HER in processing local auditory irregularities might be predominant for distinguishing between MCS and UWS. This notion is further supported by the higher accuracy of the linear discriminant classifier in classifying MCS and UWS patients by using ERP local and both HER global and local effects, as compared to all other possible feature combinations. Nonetheless, the inclusion of global effects marginally improved the classification performance, indicating that although the global effects are weaker than the local effects, they might provide complementary information to the local effects.

Our results contribute to the extensive experimental evidence showing that brain-heart interactions, as measured with HERs, are related to perceptual awareness (*Azzalini et al., 2019*; *Skora et al., 2022*). For instance, neural responses to heartbeats correlate with perception in a visual detection task (*Park et al., 2014*). Further evidence exists on somatosensory perception, where a higher detection of somatosensory stimuli occurs when the cardiac cycle is in diastole and it is reflected in

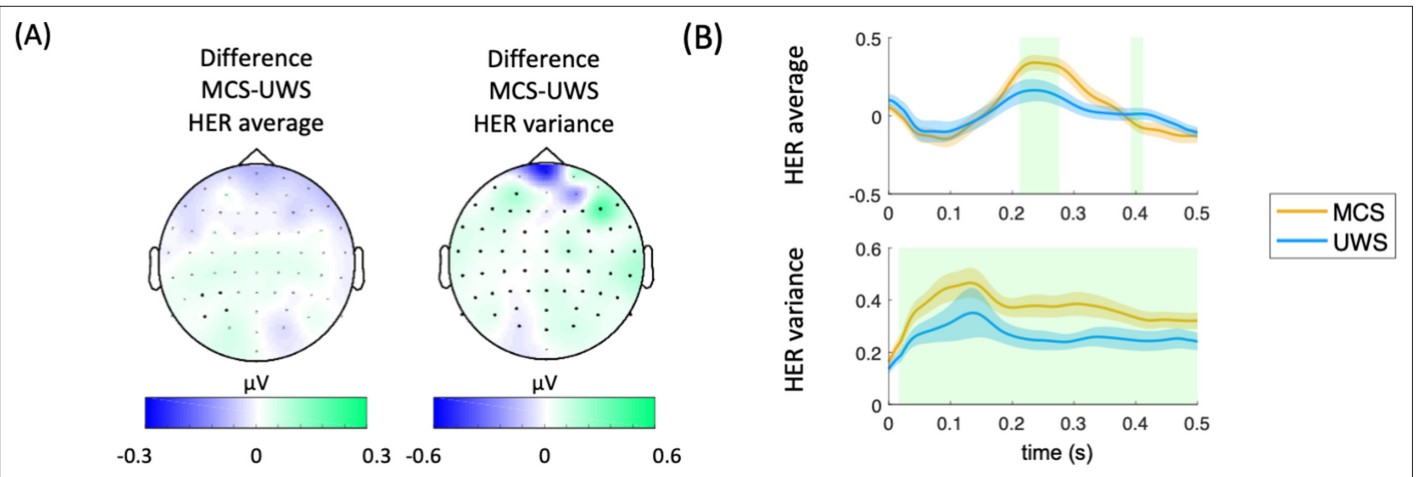

**Figure 5.** Results on HER average and HER variance for the whole protocol. (**A**) HER scalp topographies of the differences between MCS and UWS patients. Thick electrodes show significant differences after cluster permutation. (**B**) HER variance in MCS and UWS patients in the significant cluster. Shaded green areas indicate the segments in which a clustered effect was found when comparing MCS and UWS groups. HERs: heartbeat-evoked responses, MCS: minimally conscious state, UWS: unresponsive wakefulness syndrome.

HERs (*Al et al., 2020*). Evidence on heart transplanted patients show that the ability of heartbeats sensation is reduced after surgery and recovered after one year, with the evolution of the heartbeats sensation recovery reflected in the neural responses to heartbeats as well (*Salamone et al., 2020*). The responses to heartbeats also covary with self-perception: bodily-self-identification of the full body (*Park et al., 2016*), and face (*Sel et al., 2017*), and the self-relatedness of spontaneous thoughts (*Babo-Rebelo et al., 2016*) and imagination (*Babo-Rebelo et al., 2019*). Moreover, brain-heart interactions measured from heart rate variability correlate with conscious auditory perception as well (*Banellis and Cruse, 2020*; *Pérez et al., 2021*; *Pfeiffer and De Lucia, 2017*).

We showed that ERPs and HERs are repeatedly larger in MCS patients, as compared to UWS, in both local and global effects. Furthermore, the ERPs and HERs (both for the local and global effects) are uncorrelated in all possible comparisons (see *Figure 4A*), in addition to the results show differentiation of clustering effects in HER and ERP (see *Figure 4B*). These results suggest that the neuronal mechanisms behind these ERPs and HERs responses are independent. In addition, we found that HER variance is higher in MCS patients than in UWS patients, as previously reported in resting state (*Candia-Rivera and Machado, 2023a*). These results suggest that there are two distinct neuronal signatures that set apart patients in a MCS from those in an UWS. A first process probed with HER variability differentiates, irrespective of the current stimulus types being processed. This first process originates from the central and right temporal scalp areas and has been linked with social cognition but could also correspond to a self-consciousness-state markers (*Candia-Rivera et al., 2021a*). Second, a modulation of HER in response to local and global auditory irregularities. These responses present several properties related to a neural signature of conscious access to local and global deviant stimuli. Such ERPs and HERs modulations by conscious access to a new stimulus attribute may well correspond to a self-consciousness updating process occurring 'downstream' to conscious access (*Sergent and Naccache, 2012*), and enabled, for instance, in a *global neuronal workspace* architecture (*Dehaene and Naccache, 2001*).

Note that outliers are expected in disorders of consciousness, and an exact physiological characterization of the different levels of consciousness remains challenging. First, the standard assessment of consciousness based on behavioral measures has shown a high rate of misdiagnosis in MCS and UWS (*Stender et al., 2014*). The cause of the misdiagnosis of consciousness arises because consciousness does not necessarily translate into overt behavior (*Hermann et al., 2021*). Unresponsive and minimally conscious patients, namely non-behavioral MCS (*Thibaut et al., 2021*), represents the main diagnostic challenge in clinical practice. Second, some of these patients suffer from conditions that may translate into no response to stimuli, even in the presence of consciousness. For instance, when they suffer from constant pain, fluctuations in arousal levels, or sensory impairments caused by brain damage (*Chennu et al., 2013*). Third, these patients were recorded in clinical setups, which may lead to a lower signal-to-noise ratio, and consecutively lead to biased measurements in evoked potentials (*Clayson et al., 2013*).

A plethora of complementary neuroimaging techniques have been proposed to enhance the consciousness diagnosis, including anatomical and functional magnetic resonance imaging and positron emission tomography (*Kondziella et al., 2020*; *Sanz et al., 2021*). However, those methodologies may not be accessible in all clinical setups, because of costs or medical contraindications. The foregoing evidence of EEG-based techniques to diagnose consciousness (*Bai et al., 2021*; *Engemann et al., 2018*) shows promising and low-cost opportunities to develop diagnostic methods that can capture residual consciousness. Our results contribute more evidence of the potential of EEG as a diagnostic tool, but also to the role of visceral signals in consciousness (*Azzalini et al., 2019*; *Candia-Rivera, 2022a*; *Sattin et al., 2020*). This study gives evidence that HERs detect auditory conscious perception, in addition to the residual signs of consciousness in the resting-state (*Candia-Rivera et al., 2021a*).

## Materials and methods
### Patients
This study includes 46 MCS, 40 UWS patients, and 11 healthy controls. Patients were admitted at the Department of Neurology, Pitié-Salpêtrière Hospital (Paris, France) for consciousness evaluation through Coma Recovery Scale-Revised (CRS-R) (*Giacino et al., 2004*).

The study was approved by the local ethics committee (Ethical committee of the French Society of Intensive Care Medicine - SRLF; Paris, France, NEURO-DoC/HAO-006/20130409, and M-NEU-RO-DoC/NCT04534777). Informed consent was signed by the patients' legal representatives for approval of participation in the study, as required by the declaration of Helsinki.

## Experimental paradigm

Patients were recorded with high-density EEG (EGI 256 channels, 250 Hz sampling rate, referenced to the vertex) under the local-global paradigm that aims to evaluate the cognitive processing of local–short-term–, and global–long-term–auditory regularities (*Figure 1A*; *Bekinschtein et al., 2009*). The paradigm consists of two embedded levels of auditory regularities in trials formed by five consecutive sounds. The 5[th] sound defines whether the trial is standard or deviant at two levels: local and global. The local level of regularity is defined within the trial. The global level of regularities is defined across trials (frequent trials ~80% define the regularity, and rare ones ~20% violate this regularity). In *Figure 1A*, in the XX blocks, the frequent stimulus corresponds to five equal sounds (local standard and global standard). In contrast, the infrequent stimulus corresponds to four equal sounds followed by a fifth different sound (local deviant and global deviant). In the XY blocks, the frequent stimulus corresponds to four equal sounds and a fifth different sound (local deviant and global standard). The infrequent stimulus corresponds to five equal sounds (local standard and global deviant). The patients included in this study performed at least four blocks (2 XX and 2 XY), in which one block has an approximate duration of 200 s. Each trial is formed by five consecutive sounds lasting 50 milliseconds, with a 150 millisecond gap between the sounds' onsets and an intertrial interval ranging from 1350–1650 milliseconds.

The healthy controls participating in this study completed two blocks of the local-global paradigm, one XX and one XY. It is important to note that they were included solely as a reference group for qualitative analyses. The purpose of including healthy controls in our study was to determine if MCS patients exhibit similar trends in markers where a differentiation between MCS and UWS/VS patients was observed.

## Data preprocessing

MATLAB and Fieldtrip toolbox were used for data processing and analysis (*Oostenveld et al., 2011*). EEG data were offline filtered with a 1–25 Hz Butterworth band-pass order four filter, with a Hamming windowing at cutoff frequencies. The channels with large artifacts were rejected based on the area under the curve of their z-score. Channels exceeding >3 standard deviations were discarded iteratively (11±1 SEM channels rejected on average). Following the procedure described in *Raimondo et al., 2017*, electrocardiograms (ECG) were recovered from the cardiac field artifact captured in EEG data using Independent Component Analysis (ICA) (default parameters from Fieldtrip). From this, ICA-corrected EEG data and an electrocardiogram derived from independent component analysis (ICA-ECG) is obtained. Note that the use of ICA-ECG instead of a standard ECG measured from the rib cage was successfully used in other two studies (*Candia-Rivera et al., 2021a*; *Raimondo et al., 2017*). Furthermore, it was shown that the differences between the R-peak timings obtained from the ECG and ICA-ECG differ in a range of 0–4 ms (*Candia-Rivera et al., 2021a*).

To identify further noisy channels, the mean weighted-by-distance correlation of all channels between their neighbors were computed (36±2 SEM channels rejected on average). Neighborhood relationships considered all channels up to distances of 4 cm. Channels with a mean weighted-by-distance correlation lower than 80% were replaced by spline interpolation of neighbors. EEG dataset was re-referenced using a common average and a subset of 64 channels were selected for data analysis (*Candia-Rivera et al., 2021b*).

Heartbeats were detected on the ICA-ECG using an automated process based on a sliding time window detecting local maxima (R-peaks). Both peak detection and resulting histograms of interbeat interval duration were visually inspected in each patient. Ectopic interbeat intervals were automatically identified for review by detecting peaks on the derivative of the interbeat intervals time series. Manual addition/removal of peaks was performed if needed (23±3 SEM manual corrections to individual heartbeats on average).

HERs (*Park and Blanke, 2019*; *Schandry et al., 1986*) were computed by averaging EEG epochs from the R-peaks that follow the 5[th] sound from all the trials, up to 500 ms (*Figure 1B*). Epochs with

amplitude larger than 300 µV on any channel, or where the next or preceding heartbeat occurred at an interval shorter than 500 ms, were discarded. The epochs in which the stimuli were located at less than 20 ms from the closest R-peaks were discarded as well. We also controlled that the average latency between the 5[th] sound and the next heartbeat did not differ between MCS and UWS patients (Wilcoxon tests, local standard: p=0.2303, Z=1.1991; local deviants: p=0.3387, Z=0.9567; global standard: p=0.2047, Z=1.2684; global deviant: p=0.4182, Z=0.8095).

Auditory event-related potentials (ERPs) were computed for contrast by averaging EEG epochs from the 5[th] sound onset from all the trials, up to 1000 ms. Epochs with amplitude larger than 300 µV on any channel were discarded.

## Data analysis

Two neural signatures were computed to compare MCS and UWS patients: ERPs, that relate to the average of EEG epochs locked to the auditory stimuli, and HERs that relate to the average of EEG epochs locked to the heartbeats that follow the auditory stimuli. The experimental conditions, in which ERPs and HERs were used to compare MCS and UWS patients, are:

- Local effect: average of the EEG epoch associated with local deviants (local deviant/global standard epochs + local deviant/global deviant epochs), minus the average of EEG epochs associated with local standards (local standard/global standard epochs + local standard/global deviant epochs).
- Global effect: average of the EEG epoch associated to global deviants (local standard/global deviant epochs + local deviant/global deviant epochs), minus the average of EEG epochs associated to global standards (local standard/global standard epochs + local deviant/global standard epochs).

Additionally, HERs average and HERs variance were analyzed during the whole experimental protocol, i.e., the neural responses to heartbeats were analyzed with respect to all heartbeats independently of stimuli.

## Statistical analysis

Statistical comparisons were based on Wilcoxon rank sum and Spearman correlation, as specified in the main text. p-values were corrected for multiple comparisons by applying the Bonferroni rule or by using cluster-permutation analyses.

Clustered effects were revealed using a non-parametric version of cluster permutation analysis (*Candia-Rivera and Valenza, 2022b*). In brief, the cluster-based permutation test included a preliminary mask definition, identification of candidate clusters, and the computation of cluster statistics with Monte Carlo's p-value correction. The preliminary mask was defined through an unpaired Wilcoxon test, with alpha = 0.05. The identification of neighbor channels were based on the default Fieldtrip channels' neighborhood definition for 64 channels. A minimum cluster size of four channels was imposed. Adjacent candidate clusters on time were wrapped if they had at least one channel in common. Cluster statistics were computed from 10,000 random partitions. The proportion of random partitions that resulted in a lower p-value than the observed one was considered as the Monte Carlo p-value, with significance at alpha = 0.05. The cluster statistic considered is the Wilcoxon's absolute maximum Z-value obtained from all the samples of the mask.

Additionally, to confirm the presence of true effects in HERs, we compared the combined clustered effects with surrogates. We reallocated each heartbeat timing using a uniformly distributed pseudorandom process, between the first and the last sample of each recording. We computed 100 surrogates and repeated the aforementioned statistical analysis. We computed Monte Carlo p-values as the proportion of the combined clustered effects found in the surrogates with a higher effect and cluster size, with respect to the real heartbeat timings.

Lastly, in order to assess the complementarity of clusters identified in ERPs and HERs across local and global effects, we employed a fivefold cross-validation to train a linear discriminant classifier (*Fisher, 1936*), as implemented in MATLAB. The accuracies and confusion matrices were reported to evaluate the performance of the features' combinations and to quantify the occurrence of 'false MCS' and 'false UWS' predictions.

## Acknowledgements

We thank all of the participants who took part in the studies. We would like to thank the work and support of the clinicians at the Neuro ICU, DMU Neurosciences, APHP- Sorbonne Université, Hôpital de la Pitié Salpêtrière, Paris, France; and the patient families whose consent and understanding are essential to the progress of the field.

## Additional information

### Funding

| Funder | Grant reference number | Author |
|---|---|---|
| Canadian Institute for Advanced Research | | Catherine Tallon-Baudry |
| Agence Nationale de la Recherche | ANR-17-EURE-0017 | Catherine Tallon-Baudry |
| Agence Nationale de la Recherche | ANR-10- IAIHU-06 | Jacobo D Sitt |
| Sorbonne Université | EMERGENCE | Jacobo D Sitt |
| European Commission | JTC2019 | Jacobo D Sitt |

The funders had no role in study design, data collection and interpretation, or the decision to submit the work for publication.

### Author contributions

Diego Candia-Rivera, Conceptualization, Formal analysis, Investigation, Methodology, Writing – original draft, Writing – review and editing; Federico Raimondo, Conceptualization, Data curation, Investigation, Methodology, Writing – review and editing; Pauline Pérez, Writing – review and editing; Lionel Naccache, Conceptualization, Supervision, Writing – review and editing; Catherine Tallon-Baudry, Conceptualization, Formal analysis, Supervision, Methodology, Writing – review and editing; Jacobo D Sitt, Conceptualization, Data curation, Formal analysis, Supervision, Funding acquisition, Methodology, Writing – review and editing

### Author ORCIDs

Diego Candia-Rivera https://orcid.org/0000-0002-4043-217X
Federico Raimondo http://orcid.org/0000-0003-4087-8259
Catherine Tallon-Baudry http://orcid.org/0000-0001-8480-5831
Jacobo D Sitt https://orcid.org/0000-0002-3878-4846

### Ethics

The study was approved by the ethics committee of CPP Île de France 1 (Paris, France). Informed consent was signed by the patients' legal representatives for approval of participation in the study, as required by the declaration of Helsinki. (NEURO-DoC/HAO-84 006/20130409 and M-NEURO-DoC/ NCT04534777).

### Decision letter and Author response

Decision letter https://doi.org/10.7554/eLife.75352.sa1
Author response https://doi.org/10.7554/eLife.75352.sa2

## Additional files

### Supplementary files

• MDAR checklist

### Data availability

The data used in this study can be made available upon reasonable request. Because of the sensitive nature of the clinical information concerning the patients, the ethics protocol does not allow open data sharing. To access the raw data, the potential interested researcher would need to contact

the corresponding authors of the study. Together they would need to ask for an authorization from the local ethics committee, CPP Île de France 1 (Paris, France). The codes and pre-processed data are available at https://github.com/diegocandiar/brain_heart_doc, (copy archived at *Candia-Rivera, 2021*).

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
