## [Editor Report]

This study shows that neural responses to sounds and to heartbeats are affected in different ways by short-term and long-term auditory irregularities in patients diagnosed with a minimally conscious state or unresponsive wakefulness syndrome. While the findings would have been more robust had the authors collected data in the same way from a larger group of control subjects, they highlight the potential value of using heartbeat-evoked responses to inform the bedside diagnosis of disorders of consciousness. More generally, they will of interest to researchers studying brain-body interactions and their relationship to perceptual awareness.

---

## [Decision Letter]

**Decision letter after peer review:**

Thank you for submitting your article "Processing of slow-global auditory regularities causes larger neural responses to heartbeats in patients under minimal consciousness state, compared to unresponsive wakefulness syndrome" for consideration by *eLife*. Your article has been reviewed by 3 peer reviewers, including Maria Chait as the Reviewing Editor and Reviewer #3, and the evaluation has been overseen by Andrew King as the Senior Editor. The following individual involved in review of your submission has agreed to reveal their identity: Marta Garrido (Reviewer #2).

Essential revisions:

We found the work potentially very interesting, however several key issues, critical for supporting the claims in this paper, were identified. We hope you can address these in a revision.

1) We were confused by the specificity of HER differences vs straightforward prediction error differences in auditory evoked responses. Is there a significant local and global deviant response in each patient group?

2) In general, the interpretation of the results seems to rely on significant results for some contrasts but on others being non-significant. Why do you not look for an interaction?

3) The pattern of results across groups (Panel E; see specific comments from Rev2 and Rev3 below) was somewhat counterintuitive and raises questions about what this neural signature can tell us about the state of consciousness.

4) We found the Discussion section to be too sparse. We encourage the authors to add a discussion of the implications of these data to our understanding of the consciousness state of MCS and UWS patients.

Please also see further comments in the individual reviews.

*Reviewer #1 (Recommendations for the authors):*

I really enjoyed reading this paper and have comments really to clarify the experimental design and results.

The design looks like it is a 2x2 design where there is a standard or deviant that is modulated globally or locally. I think it is more complicated than that and I am happy to be wrong on this. The reason for mentioning it is that the analysis focuses on the main effects of global and local and not on the interaction between them. This seems important as the main result here is a significant effect for global and a non significant effect for local. It would be great if this could be shown as a significant interaction. If the design can is not a 2x2 maybe the authors could explain this in the manuscript to help those like me who conceptually struggled with the design.

The long term goal as described is to be able to use EEG as a diagnostic tool for post-comatose patients. It would be really helpful for the more general reader to have a clearer understanding of where this study fits in this translational pipeline.

*Reviewer #2 (Recommendations for the authors):*

Methodology:

1) ECG signals were determined using ICA on scalp EEG rather than actual electrodes on the chest? I suggest including some discussion on the limitations with this approach – how accurate is this approach relatively to ECG?

2) It appears that the addition and removal of peaks was performed manually? If so, there is room for subjectivity with this approach making it hard to replicate/reproduce. Can detailed information about which trials etc were excluded be provided together with the dataset, such that these results can be reproduced in the future?

3) The controls sample (N=11) is relatively quite small when compared to N=59 UWS and N=58 MCS.

Results:

4) Figure 1E – could the effect observed driven by the 4 MCS participants that appear to be outliers? I think this is important to check particularly as it seems that UWS is more similar to EMCS and the Healthy group than EMCS, which is counterintuitive – is that really the case? If so, why would that be?

5) Related to the point above, it would be helpful to know how the variance of HER looks for the other 2 groups (EMCS and Healthy). Is there a relationship between HER variance and consciousness? If not, how to interpret this result?

Discussion:

6) I was confused about the issues of specificity of HER differences vs straightforward prediction error differences in auditory evoked responses locked to the 5th sound. The results state the following: "Group-wise HER average differs from EEG activity locked to the 5th sound, as compared in the same cluster or the best cluster (MANOVA test, χ 2 153=16.6366, df=3, value=0.0008)". This seems to reflect the data displayed in Figure 1D third row? (if so, an asterisk could be placed to clearly indicate differences group differences). However, this is not well discussed. It is said: "Various tests showed that locking EEG to heartbeats is necessary to find distinctions between patients, revealing the existence of a possible auditory-cardiac synchrony 207 (Pérez et al., 2021; Pfeiffer and Lucia, 2017). This study gives evidence that HERs detect auditory conscious perception, in addition to residual signs of consciousness in resting state 209 (Candia-Rivera et al., 2021a)." I agree that this study shows that locking EEG to heartbeats shows a difference between UWC and MCS, but it doesn't seem to show that it is necessary given the statistical difference reported for EEG locked to the 5th sound (best cluster). This should be clarified and discussed. What are the implications if it's not specific?

7) Links are made to studies on social cognition and the global workspace theory but I couldn't see how these data really speak to those theories. The abstract clearly states how this work could be useful in informing diagnosis of the state of consciousness, which is in keeping with the data, and yet the discussion glosses over that.

8) How do the authors interpret the fact that the HER effect is present for global but nor for local deviants?

*Reviewer #3 (Recommendations for the authors):*

I found the Discussion section to be too sparse, especially for a general Neuro journal like *eLife*. I encourage the authors to add a discussion of the implications of these data to our understanding of the consciousness state of MCS and UWS patients. Similarly, the intro should have motivated the specific paradigm used here (local-global) in more detail. Why is this a particularly useful stimulus?

From an applied perspective – Effects were perhaps present at the group level but the patients show so much variability (more than the two control groups..)….what is the vision for using an approach like this to diagnose consciousness?

[Editors' note: further revisions were suggested prior to acceptance, as described below.]

Thank you for resubmitting your work entitled "Conscious processing of global and local auditory irregularities causes differentiated heartbeat-evoked responses" for further consideration by *eLife*. Your revised article has been evaluated by Andrew King (Senior Editor) and a Reviewing Editor.

The manuscript has been improved but there are key remaining issues that need to be addressed, as outlined below:

1) The post-hoc decision to remove the control data is worrying, especially since the data we originally saw raised some concerns that undermined the main conclusions in the manuscript. The justification for setting the exclusion criteria as they were is not very clear, and instead, it is preferable for the control group data to be improved (e.g. by acquiring appropriate data) and re-included.

2) Additionally, it is difficult to evaluate the nature of any effects since no scalp level (time domain) data are being presented.

3) Please also see comments from Reviewer 2 below regarding clarification and interpretation.

*Reviewer #2 (Recommendations for the authors):*

This paper investigates the ERPs and HERs evoked in a global/local MMN paradigm in MCS and UWS patients. The critical claim is that local and global effects produce distinct ERPs and HERs for the two groups of patients, which can help distinguish between them and hence improve diagnosis.

It would be really important to understand how much HERs improve diagnosis above and beyond ERPs. If so, this would be potentially very impactful in clinical practice. By eye-balling 2B, 3B, and 4A, it seems that it might be the case but we'd need some quantification for how much more that is to be fully convinced.

I thought that the description of the results lacked some context and interpretation. More specifically, I was hoping to see more of "in order to test hypothesis X we performed the following analysis… We found Y which means Z". This applies to all results figures.

In Figures 2B and 3B, it's unclear what exactly is meant by "the clustered effects were combined into a single value" – can the authors clarify what the computation is? Averaged across time points, or something else?

"The local effect, as evaluated with HERs, showed a better separability between MCS and UWS patients, and a greater specificity with respect to surrogate heartbeat analysis." Unclear what is the data/figure/analysis that supports this claim – table 2, Figure 4D? Suggest making this explicit either in the discussion or results.

"Nevertheless, the measured responses in ERPs and HERs do not separate MCS and UWS patients' groups completely – do we have a quantification of the degree of overlap?" Figure 4 is illustrative but it would be good to have some sort of quantification.

*Reviewer #3 (Recommendations for the authors):*

This revised manuscript differs substantially from the originally submitted manuscript. A Post Hoc inclusion criterion imposed on the data resulted in the removal of the control group altogether. I do not think this is correct to do. That group was important to allow one to compare responses in patients relative to controls. The appropriate course of action, if it is deemed that original data were not of sufficient quality is to collect more data.

This, and the fact that ERP responses are not plotted (e.g. in Figure 2, 3) make it very difficult to interpret findings.

I am not an expert in heart-brain interactions (but am an expert in EEG/auditory oddball responses) and found it very difficult to evaluate the results without seeing the actual EEG time-domain data.

---

## [Author Response]

Essential revisions:We found the work potentially very interesting, however several key issues, critical for supporting the claims in this paper, were identified. We hope you can address these in a revision.1) We were confused by the specificity of HER differences vs straightforward prediction error differences in auditory evoked responses. Is there a significant local and global deviant response in each patient group?2) In general, the interpretation of the results seems to rely on significant results for some contrasts but on others being non-significant. Why do you not look for an interaction?3) The pattern of results across groups (Panel E; see specific comments from Rev2 and Rev3 below) was somewhat counterintuitive and raises questions about what this neural signature can tell us about the state of consciousness.4) We found the Discussion section to be too sparse. We encourage the authors to add a discussion of the implications of these data to our understanding of the consciousness state of MCS and UWS patients.Please also see further comments in the individual reviews.

Since one of the main concerns from reviewers and the editor was that healthy participants did not show an effect to the global effect, we realized that the previous analysis included participants where only a very limited amount of data was obtained. We now (properly) apply an inclusion criteria specifying that participants’ data should include at least 4 blocks (2 XX and 2 XY; Figure 1). With this new criterion, the new sets are 46 MCS and 40 UWS. The specific response to editor’s points, as follows:

(1) Based on our new results. Yes, there is a differentiated response to local and global effects, in both ERPs and HERs. MCS patients present repeatedly a larger response in all the comparisons.

(2) We have performed a simple correlation analysis to show that there is a distinction with respect to local and global deviants.

(3) The absence of correlates of consciousness in healthy participants (and EMCS) was due to the inclusion of participants’ data which were way too small to be reliable. Therefore, the new inclusion criteria did not include any EMCS or healthy participants.

(4) We have included a more comprehensive discussion in this revision, including the most recent findings on brain-heart interactions and conscious processing of exteroceptive information.

Reviewer #1 (Recommendations for the authors):I really enjoyed reading this paper and have comments really to clarify the experimental design and results.The design looks like it is a 2x2 design where there is a standard or deviant that is modulated globally or locally. I think it is more complicated than that and I am happy to be wrong on this. The reason for mentioning it is that the analysis focuses on the main effects of global and local and not on the interaction between them. This seems important as the main result here is a significant effect for global and a non significant effect for local. It would be great if this could be shown as a significant interaction. If the design can is not a 2x2 maybe the authors could explain this in the manuscript to help those like me who conceptually struggled with the design.

We thank the reviewer for pointing to the data analysis description, which may allow future readers for a better understanding of the study design.

The comparisons done are between MCS and UWS patients for the following conditions:

Auditory evoked potential related to the local effect (contrast of local deviant versus local standard EEG activity locked to the auditory stimuli)Auditory evoked potential related to the global effect (contrast of global deviant versus local standard EEG activity locked to the auditory stimuli)Heartbeat evoked potential related to the local effect (contrast of local deviant versus local standard EEG activity locked to the heart R-peak)Heartbeat evoked potential related to the global effect (contrast of global deviant versus local standard EEG activity locked to the heart R-peak)

We have modified the Materials and methods to clarify on that:

“Data analysis

Two neural signatures were computed to compare MCS and UWS patients: ERPs, that relate to the average of EEG epochs locked to the auditory stimuli, and HERs that relate to the average of EEG epochs locked to the heartbeats that follow the auditory stimuli. The experimental conditions, in which ERPs and HERs were used to compare MCS and UWS patients, are:

Local effect: average of the EEG epoch associated to local deviants (local deviant/global standard epochs + local deviant/global deviant epochs), minus the average of EEG epochs associated to local standards (local standard/global standard epochs + local standard/global deviant epochs).

Global effect: average of the EEG epoch associated to global deviants (local standard/global deviant epochs + local deviant/global deviant epochs), minus the average of EEG epochs associated to global standards (local standard/global standard epochs + local deviant/global standard epochs).

Additionally, HERs average and HERs variance were analyzed during the whole experimental protocol, i.e., the neural responses to heartbeats were analyzed with respect all heartbeats independently of stimuli.”

We have split those results in two new figures (2 and 3) to clarify on the differences between local and global effects.

The long term goal as described is to be able to use EEG as a diagnostic tool for post-comatose patients. It would be really helpful for the more general reader to have a clearer understanding of where this study fits in this translational pipeline.

We have included in the discussion a paragraph explaining the potential contribution of our work in the clinical practice:

“A plethora of complementary neuroimaging techniques have been proposed to enhance the consciousness diagnosis, including anatomical and functional magnetic resonance imaging and positron emission tomography (Kondziella et al., 2020; Sanz et al., 2021). However, those methodologies may not be accessible in all clinical setups, because of costs or medical contraindications. The foregoing evidence of EEG-based techniques to diagnose consciousness (Bai et al., 2021; Engemann et al., 2018) shows promising and low-cost opportunities to develop diagnostic methods that can capture residual consciousness. Our results contribute more evidence of the potential of EEG as a diagnostic tool, but also to the role of visceral signals in consciousness (Azzalini et al., 2019; Candia-Rivera, 2022; Sattin et al., 2020). This study gives evidence that HERs detect auditory conscious perception, in addition to the residual signs of consciousness in resting-state (Candia-Rivera et al., 2021a).”

Reviewer #2 (Recommendations for the authors):Methodology:1) ECG signals were determined using ICA on scalp EEG rather than actual electrodes on the chest? I suggest including some discussion on the limitations with this approach – how accurate is this approach relatively to ECG?

We have included a new paragraph in Materials and methods section to clarify on the reliability of using ICA-ECG when the ECG is not available: “From this, ICA-corrected EEG data and an electrocardiogram derived from independent component analysis (ICA-ECG) is obtained. Note that the use of ICA-ECG instead of a standard ECG measured from the rib cage was successfully used in other two studies (Candia-Rivera et al., 2021a; Raimondo et al., 2017). Furthermore, it was shown that the differences between the R-peak timings obtained from the ECG and ICA-ECG differ in a range of 0-4 ms (Candia-Rivera et al., 2021a).”

2) It appears that the addition and removal of peaks was performed manually? If so, there is room for subjectivity with this approach making it hard to replicate/reproduce. Can detailed information about which trials etc were excluded be provided together with the dataset, such that these results can be reproduced in the future?

The methodology does not rely on the user capacity to identify R-peaks. The procedure of R-peak detection is performed by an automatic R-peak template matching method, which is reliable and tested in several other studies.

The detection of ectopic interbeat intervals was performed automatically as well, by detecting peaks on the derivative of the heart rate. As a final step, a visual overview of the automatically detected ectopic intervals is performed for manual corrections.

We have modified the Materials and methods section to clarify on this point:

“Heartbeats were detected on the ICA-ECG using an automated process based on a sliding time window detecting local maxima (R-peaks). Both peak detection and resulting histogram of interbeat interval duration were visually inspected in each patient. Ectopic interbeat intervals were automatically identified for review by detecting peaks on the derivative of the interbeat intervals time series. Manual addition/removal of peaks was performed if needed (23 ± 3 SEM manual corrections to individual heartbeats on average).”

3) The controls sample (N=11) is relatively quite small when compared to N=59 UWS and N=58 MCS.

The new inclusion criteria did not include any of the 11 controls because of not enough trials available for those participants.

Results:4) Figure 1E – could the effect observed driven by the 4 MCS participants that appear to be outliers? I think this is important to check particularly as it seems that UWS is more similar to EMCS and the Healthy group than EMCS, which is counterintuitive – is that really the case? If so, why would that be?

Since this revision included a different group of patients. we have performed the analyses again, including a figure showing the distribution of the clustered effects found, to show there is a clear group trend in MCS patients to present overall larger responses (ERPs and HERs).However, outliers are expected in patients with disorders of consciousness. We have included a paragraph in the discussion about this:

“Note that outliers are expected in disorders of consciousness and exact physiological characterization of the different levels of consciousness remains challenging. First, the standard assessment of consciousness based on behavioral measures has shown a high rate of misdiagnosis in MCS and UWS (Stender et al., 2014). The cause of the misdiagnosis of consciousness arises because consciousness does not necessarily translate into overt behavior (Hermann et al., 2021). Unresponsive and minimally conscious patients, namely non-behavioral MCS (Thibaut et al., 2021), represent the main diagnostic challenge in clinical practice. Second, some of these patients suffer from conditions that may translate to no response to stimuli, even in presence of consciousness. For instance, when they suffer from constant pain, fluctuations in arousal levels, or sensory impairments caused by brain damage (Chennu et al., 2013). Third, these patients were recorded in clinical setups, which may lead to a lower signal-to-noise ratio, and lead to biased measurements in evoked potentials (Clayson et al., 2013).”

5) Related to the point above, it would be helpful to know how the variance of HER looks for the other 2 groups (EMCS and Healthy). Is there a relationship between HER variance and consciousness? If not, how to interpret this result?

It was shown in Candia-Rivera et al., 2021 (J Neurosci) in an independent cohort of UWS and MCS patients that HER variance in the right hemisphere contributed to the classification of these patients up to 87% accuracy. Importantly, HER variance in that specific scalp region do not separate by itself patients’ groups, and it rather contribute to all features used in the automatic classifier (HER average and variance in all channels at different timings between 200-400 ms). In this study we show that HER variance also may contribute to distinguish UWS and MCS patients. Likewise, that measure cannot fully separate these patients’ groups in this cohort.

Note that HER variance is a measure not explored before Candia-Rivera et al., 2021. Therefore, no further evidence exists on this marker in healthy participants, nor other paradigms. Given the sample size limitation of these groups in the initial inclusion criteria: EMCS (n=20) and healthy (n=10), and the new inclusion criteria did not include EMCS and healthy participants, we did not explore further differences on HER variance on those participants.

Discussion:6) I was confused about the issues of specificity of HER differences vs straightforward prediction error differences in auditory evoked responses locked to the 5th sound. The results state the following: "Group-wise HER average differs from EEG activity locked to the 5th sound, as compared in the same cluster or the best cluster (MANOVA test, χ 2 153=16.6366, df=3, value=0.0008)". This seems to reflect the data displayed in Figure 1D third row? (if so, an asterisk could be placed to clearly indicate differences group differences). However, this is not well discussed. It is said: "Various tests showed that locking EEG to heartbeats is necessary to find distinctions between patients, revealing the existence of a possible auditory-cardiac synchrony 207 (Pérez et al., 2021; Pfeiffer and Lucia, 2017). This study gives evidence that HERs detect auditory conscious perception, in addition to residual signs of consciousness in resting state 209 (Candia-Rivera et al., 2021a)." I agree that this study shows that locking EEG to heartbeats shows a difference between UWC and MCS, but it doesn't seem to show that it is necessary given the statistical difference reported for EEG locked to the 5th sound (best cluster). This should be clarified and discussed. What are the implications if it's not specific?

We thank the reviewer for pointing to the interpretation of the results and statistics.

We have re-run all analyses and we have kept only a correlation analysis to show that the different markers may likely represent different cognitive processes that contribute to the consciousness assessment.

7) Links are made to studies on social cognition and the global workspace theory but I couldn't see how these data really speak to those theories. The abstract clearly states how this work could be useful in informing diagnosis of the state of consciousness, which is in keeping with the data, and yet the discussion glosses over that.

We have included in the introduction the motivation of applying the local-global paradigm:

“We hypothesized that HERs can be modulated by contextual processing of different levels of auditory regularities, as presented in the local-global paradigm (Bekinschtein et al., 2009). In this study, we analyze HERs following the presentation of auditory irregularities, with special regard on distinguishing UWS (n=40) and MCS (n=46) patients. Note that the automated classification of this cohort was previously performed in another study (Raimondo et al., 2017). Therefore, our aim is to characterize the group-wise differences between UWS and MCS patients that may allow a multi-dimensional cognitive evaluation to infer the presence of consciousness (Sergent et al., 2017), but also complement the bedside diagnosis performed with neuroimaging methods that capture neural correlates of covert consciousness (Sanz et al., 2021).”

8) How do the authors interpret the fact that the HER effect is present for global but nor for local deviants?

Indeed, with the new analyses we now show that both, local and global responses exist. We have added in the discussion the interpretation of the results:

“We showed that ERPs and HERs are repeatedly larger in MCS patients, as compared to UWS, in both local and global effects. Furthermore, the ERPs and HERs (both for the local and global effects) are uncorrelated in all possible comparisons (see Figure 4A), in addition to the results showing differentiation of clustering effects in HER and ERP (see Figure 4B). These results suggest that the neuronal mechanisms behind these ERPs and HERs responses are independent. In addition, we found that HER variance is higher in MCS patients than in UWS patients, as previously reported in resting state (Candia-Rivera et al., 2021a). Put together these results suggest that two different neuronal signatures differentiate MCS from UWS patients. A first process probed with HER variability differentiates, irrespective of the current stimulus type being processed. This first process originates from central and right temporal scalp areas and has been linked with social cognition but could also correspond to a self-consciousness-state marker (Candia-Rivera et al., 2021a). Second, a modulation of HER in response to local and global auditory irregularities. These responses present several properties related to a neural signature of conscious access to local and global deviant stimuli. Such ERPs and HERs modulations by conscious access to a new stimulus attribute may well correspond to a self-consciousness updating process occurring ‘downstream’ to conscious access (Sergent and Naccache, 2012), and enabled for instance in a global neuronal workspace architecture (Dehaene and Naccache, 2001).”

Reviewer #3 (Recommendations for the authors):I found the Discussion section to be too sparse, especially for a general Neuro journal like eLife. I encourage the authors to add a discussion of the implications of these data to our understanding of the consciousness state of MCS and UWS patients. Similarly, the intro should have motivated the specific paradigm used here (local-global) in more detail. Why is this a particularly useful stimulus?From an applied perspective – Effects were perhaps present at the group level but the patients show so much variability (more than the two control groups..)….what is the vision for using an approach like this to diagnose consciousness?

Thanks for this recommendation. We have enriched the discussion in this new revision.

We have included a commentary on the potential of the local-global paradigm:

“We showed that ERPs and HERs are repeatedly larger in MCS patients, as compared to UWS, in both local and global effects. Furthermore, the ERPs and HERs (both for the local and global effects) are uncorrelated in all possible comparisons (see Figure 4A), in addition to the results showing differentiation of clustering effects in HER and ERP (see Figure 4B). These results suggest that the neuronal mechanisms behind these ERPs and HERs responses are independent. In addition, we found that HER variance is higher in MCS patients than in UWS patients, as previously reported in resting state (Candia-Rivera et al., 2021a). Put together these results suggest that two different neuronal signatures differentiate MCS from UWS patients. A first process probed with HER variability differentiates, irrespective of the current stimulus type being processed. This first process originates from central and right temporal scalp areas and has been linked with social cognition but could also correspond to a self-consciousness-state marker (Candia-Rivera et al., 2021a). Second, a modulation of HER in response to local and global auditory irregularities. These responses present several properties related to a neural signature of conscious access to local and global deviant stimuli. Such ERPs and HERs modulations by conscious access to a new stimulus attribute may well correspond to a self-consciousness updating process occurring ‘downstream’ to conscious access (Sergent and Naccache, 2012), and enabled for instance in a global neuronal workspace architecture (Dehaene and Naccache, 2001).”

We have included a commentary on where our results stand within the brain-heart interaction literature:

“Our results contribute to the extensive experimental evidence showing that brain-heart interactions, as measured with HERs, are related to perceptual awareness (Azzalini et al., 2019; Skora et al., 2022). For instance, neural responses to heartbeats correlate with perception in a visual detection task (Park et al., 2014). Further evidence exists on somatosensory perception, where a higher detection of somatosensory stimuli occurs when the cardiac cycle is in diastole and it is reflected in HERs (Al et al., 2020). Evidence on heart transplanted patients shows that the ability of heartbeats sensation is reduced after surgery and recovered after one year, with the evolution of the heartbeats sensation recovery reflected in the neural responses to heartbeats as well (Salamone et al., 2020). The responses to heartbeats also covary with self-perception: bodily-self-identification of the full body (Park et al., 2016), and face (Sel et al., 2017), and the self-relatedness of spontaneous thoughts (Babo-Rebelo et al., 2016) and imagination (Babo-Rebelo et al., 2019). Moreover, brain-heart interactions measured from heart rate variability correlate with conscious auditory perception as well (Banellis and Cruse, 2020; Pérez et al., 2021; Pfeiffer and Lucia, 2017).”

We have included a commentary on the clinical impact of our findings:

“A plethora of complementary neuroimaging techniques have been proposed to enhance the consciousness diagnosis, including anatomical and functional magnetic resonance imaging and positron emission tomography (Kondziella et al., 2020; Sanz et al., 2021). However, those methodologies may not be accessible in all clinical setups, because of costs or medical contraindications. The foregoing evidence of EEG-based techniques to diagnose consciousness (Bai et al., 2021; Engemann et al., 2018) shows promising and low-cost opportunities to develop diagnostic methods that can capture residual consciousness. Our results contribute more evidence of the potential of EEG as a diagnostic tool, but also to the role of visceral signals in consciousness (Azzalini et al., 2019; Candia-Rivera, 2022; Sattin et al., 2020). This study gives evidence that HERs detect auditory conscious perception, in addition to the residual signs of consciousness in resting-state (Candia-Rivera et al., 2021a).”

[Editors' note: further revisions were suggested prior to acceptance, as described below.]

The manuscript has been improved but there are key remaining issues that need to be addressed, as outlined below:1) The post-hoc decision to remove the control data is worrying, especially since the data we originally saw raised some concerns that undermined the main conclusions in the manuscript. The justification for setting the exclusion criteria as they were is not very clear, and instead, it is preferable for the control group data to be improved (e.g. by acquiring appropriate data) and re-included.2) Additionally, it is difficult to evaluate the nature of any effects since no scalp level (time domain) data are being presented.3) Please also see comments from Reviewer 2 below regarding clarification and interpretation.

We appreciate the editor's comments and acknowledge that it would be ideal validating our results with a larger cohort. In the first revision, we recognized that the unrestricted inclusion criteria inadvertently compromised the statistical power of our study. Consequently, the results did not align with our expectations, particularly in relation to the healthy controls. This heterogeneity within the statistical analysis potentially led to an overestimation of the observed effects.

To address these issues, we decided to narrow our focus in the second version of this manuscript. We specifically targeted the population of interest, allowing us to investigate the underlying factors that differentiate MCS and UWS/VS patients. By refining the inclusion criteria, we were able to establish a clearer contrast between the two patients’ groups. This refinement facilitated a more accurate assessment of the impact of auditory irregularities on the measures of brain-heart interaction that were studied.

We understand the concerns raised about the exclusion of the healthy control group. In this revision, we have re-included them, but solely for qualitative analysis purposes. Our intention was to address the limitation of the current population by demonstrating that the new results obtained from comparing MCS and UWS/VS patients exhibit more reliable differences, as MCS generally align with the expected trends observed in healthy controls.

We fully acknowledge the potential concerns raised regarding the exclusion and subsequent re-inclusion of the healthy control group. In order to justify this decision, we would like to provide a detailed explanation of how it strengthens the robustness and validity of our findings.

Furthermore, we would like to stress that due to current circumstances, acquiring new data has become very challenging, particularly because some of the authors are no longer affiliated with the laboratory where this study was conducted.

Considering these justifications, we kindly request that you consider our rationale for including healthy controls in the study analysis during the second revision of our manuscripts. We have prepared a revised version of the manuscript that incorporates the initial 11 healthy controls for qualitative analysis. We kindly request the editor and reviewers to evaluate our revised proposal, which we believe adequately addresses your concerns regarding our results. We believe that this decision significantly contributes to the methodological soundness and validity of our research.

Thank you for your time and consideration. We appreciated your valuable insights and feedback, which undoubtedly contributed to improving the quality of our study. Please find our responses to the specific comments from the reviewers below. The first and second points are further addressed in our responses to reviewer 3.

Reviewer #2 (Recommendations for the authors):This paper investigates the ERPs and HERs evoked in a global/local MMN paradigm in MCS and UWS patients. The critical claim is that local and global effects produce distinct ERPs and HERs for the two groups of patients, which can help distinguish between them and hence improve diagnosis.It would be really important to understand how much HERs improve diagnosis above and beyond ERPs. If so, this would be potentially very impactful in clinical practice. By eye-balling 2B, 3B, and 4A, it seems that it might be the case but we'd need some quantification for how much more that is to be fully convinced.

We thank the reviewer for their comments. We have included a panel C in Figure 4 a new analysis performing a linear discriminant classifier to quantify the complementarity of the ERPs and HERs

I thought that the description of the results lacked some context and interpretation. More specifically, I was hoping to see more of "in order to test hypothesis X we performed the following analysis… We found Y which means Z". This applies to all results figures.

We have included a short paragraph to explicitly state the objective of the study:

“In this study, we examined the cognitive processing of auditory irregularities in patients with disorders of consciousness using the local-global paradigm. This paradigm assesses the processing of both local (short-term) and global (long-term) auditory regularities. Our objective was to identify physiological responses that could differentiate between patients in MCS and those in UWS. We hypothesized that assessing auditory irregularities at both local and global levels could offer valuable insights into the distinction of MCS and UWS patients. Furthermore, this distinction may be further improved by analyzing the physiological modulation of auditory processing in relation to measures of brain-heart interactions. To achieve this, we conducted tests to investigate the local and global effects in ERPs, which involved analyzing the standard average of EEG epochs aligned with the occurrence of auditory deviants. Additionally, we explored the HERs to assess whether the neural responses to heartbeats, within the context of auditory irregularity processing, could serve as novel differentiating factors between MCS and UWS patients.”

In Figures 2B and 3B, it's unclear what exactly is meant by "the clustered effects were combined into a single value" – can the authors clarify what the computation is? Averaged across time points, or something else?

We thank the reviewer for this comment. We have included a description of the averaging procedure, as follows:

“The clustered effects were combined to obtain a single value for each patient, corresponding to ERP global and local effects. To combine the clustered effects, we computed the average of all points (channel x time) identified in the cluster permutation analysis, which effectively distinguished between patients diagnosed with MCS and UWS.”

"The local effect, as evaluated with HERs, showed a better separability between MCS and UWS patients, and a greater specificity with respect to surrogate heartbeat analysis." Unclear what is the data/figure/analysis that supports this claim – table 2, Figure 4D? Suggest making this explicit either in the discussion or results.

We thank the reviewer for this comment. We have included a description of that result, as follows:

“It is worth noting that the HER local effect demonstrated higher specificity compared to the HER global effect during the permutation test. Only 3% of randomly timed surrogate heartbeats exhibited separability that surpassed what was observed with the original heartbeats.”

"Nevertheless, the measured responses in ERPs and HERs do not separate MCS and UWS patients' groups completely – do we have a quantification of the degree of overlap?" Figure 4 is illustrative but it would be good to have some sort of quantification.

We thank the reviewer for this comment. As mentioned above, we included a new analysis on patients’ classification. The overlap in classifications is shown in the confusion matrices in Figure 4C.

Reviewer #4 (Recommendations for the authors):This revised manuscript differs substantially from the originally submitted manuscript. A Post Hoc inclusion criterion imposed on the data resulted in the removal of the control group altogether. I do not think this is correct to do. That group was important to allow one to compare responses in patients relative to controls. The appropriate course of action, if it is deemed that original data were not of sufficient quality is to collect more data.

We appreciate the reviewer's comment. We have made a few modifications to answer to the reviewer’s concerns.

In the new figures, we have incorporated the initial 11 healthy controls as a reference group. Although the healthy controls had fewer trials compared to the patients, we included them to provide context based on the new inclusion criteria. We have clarified this in the participants description as follows:

“The healthy controls participating in this study completed two blocks of the local-global paradigm, one XX and one XY. It is important to note that they were included solely as a reference group for qualitative analyses. The purpose of including healthy controls in our study was to determine if MCS patients exhibit similar trends in markers where a differentiation between MCS and UWS/VS patients was observed.”

In the previous version of the manuscript, the healthy controls did not exhibit the expected trend in the measured HERs. However, with the analysis conducted using the new inclusion criteria, we observed different clustered effects between the comparisons of MCS and UWS, which also affected the trend observed in the healthy controls. In this revised version of the manuscript, we have included again the distributions of the healthy controls as a reference point.

Our findings demonstrate that healthy controls exhibit a similar trend to MCS patients in ERP local, HER global, and HER local. However, the ERP global effect was the only feature that deviated from this trend, as discussed below:

“Furthermore, it is worth noting that the ERP global effect observed in healthy controls did not follow the same trend than MCS patients. These findings may require of further future explorations to determine if the observed effect is exclusive to MCS patients or if healthy controls do not exhibit the same effect due to their lower number of trials performed in the local-global paradigm during this study. However, the response observed in healthy controls does resemble a standard P300 response. These findings align with previous reports indicating that the ERPs during local deviants exhibit superior discriminatory ability between MCS and UWS patients (Faugeras et al., 2012). Furthermore, these results suggest that both ERP and HER in processing local auditory irregularities might be predominant for distinguishing between MCS and UWS. This notion is further supported by the higher accuracy of the linear discriminant classifier in classifying MCS and UWS patients by using ERP local and both HER global and local effects, as compared to all other possible feature combinations. Nonetheless, the inclusion of global effects marginally improved the classification performance, indicating that although the global effects are weaker than the local effects, they might provide complementary information to the local effects.”

This, and the fact that ERP responses are not plotted (e.g. in Figure 2, 3) make it very difficult to interpret findings.I am not an expert in heart-brain interactions (but am an expert in EEG/auditory oddball responses) and found it very difficult to evaluate the results without seeing the actual EEG time-domain data.

We thank the reviewer for this comment. As shown above, we have included a new panel C in the figures 2 and 3 with the time-varying responses for ERPs and HERs.